# The Origin and Maintenance of Tuberculosis Is Explained by the Induction of Smear-Negative Disease in the Paleolithic

**DOI:** 10.3390/pathogens11030366

**Published:** 2022-03-17

**Authors:** Pere-Joan Cardona, Martí Català, Clara Prats

**Affiliations:** 1Unitat de Tuberculosi Experimental, Germans Trias i Pujol Research Institute (IGTP), 08916 Badalona, Spain; 2Microbiology Department, North Metropolitan Clinical Laboratory, ‘Germans Trias i Pujol’ University Hospital, 08916 Badalona, Spain; 3Genetics and Microbiology Department, Universitat Autònoma de Barcelona, 08916 Barcelona, Spain; 4Centro de Investigación Biomédica en Red de Enfermedades Respiratorias (CIBERES), 28029 Madrid, Spain; 5Comparative Medicine and Bioimage Centre of Catalonia (CMCiB), Germans Trias i Pujol Research Institute (IGTP), 08916 Badalona, Spain; mcatala@igtp.cat (M.C.); clara.prats@upc.edu (C.P.); 6Departament de Física, Escola d’Enginyeria Agroalimentària i de Biosistemes de Barcelona, Universitat Politècnica de Catalunya (UPC)-BarcelonaTech, 08916 Badalona, Spain

**Keywords:** *Homo sapiens*, *Mycobacterium tuberculosis*, disease spectrum, Paleolithic, Neolithic, demography, resistance, tolerance, chronicity, SEIR model, coinfection, coevolution, eco-immunology, mutualism, inequality, poverty

## Abstract

Is it possible that the origin of *Mycobacterium tuberculosis* (Mtb) infection was around 70,000 years before the common era? At that time *Homo sapiens* was just another primate species with discrete growth and a very low-density geographic occupation. Therefore, it is difficult to understand the origin of a highly virulent obligate human pathogen. We have designed a new SEIR model (TBSpectr) that allows the differentiation of smear-positive and -negative tuberculosis. The model reconciles currently accepted growth rates for the Middle Paleolithic (0.003%/year) and Neolithic (0.1%/year). The obtained data link the origin of Mtb infection in the Middle Paleolithic to the induction of smear-negative TB, and reveal that its persistence required interrelations among hunter–gatherer groups, while the risk of human extinction was negligible. It also highlights the number of people infected per case and the fast progression to disease for Mtb infection maintenance, as well as the link between poor health in the Neolithic with the increased incidence of more severe forms of TB (smear-positive). In conclusion, our data support the origin of TB as a well-tolerated, highly persistent disease, even in low-density populations, showing the difficulty of its eradication and highlighting the necessity for providing better health conditions to humans to reduce its severity.

## 1. Introduction

There is a current hypothesis that establishes the origin of *Mycobacterium tuberculosis* (Mtb) infection in the middle Palaeolithic age, 70,000 years before the common era (BCE), as determined by molecular timing bases [1]. This hypothesis presents a great challenge, as it is difficult to reconcile how a pathogen with such extraordinary virulence could coevolve with a host, *Homo sapiens*, which at that time represented a fragile animal species [2,3]. In fact, tuberculosis (TB) is the greatest killer of humankind. It has been estimated to have caused 1,000,000,000 deaths in the last 200 years [4]. This impact appears to be the final phase of a formidable incidence, records of which first started in Europe in the eighteenth century, and coinciding with the industrial revolution and the compilation of the first consistent epidemiological records, with mortality incidences in big cities such as Stockholm, Hamburg and London peaking at around 900 deaths/100,000 inhabitants [5]. When the origin of this epidemic seemed to be uncertain [6], and it was thought to be the consequence of the sudden emergence of crowded cities [7], growing evidence began to emerge that in reality it had always been with us [8]. The reason for the perception of this sudden gap in the European TB incidence may lie in the lack of availability of a precise means of diagnosis until the studies by René Laennec. His work was instrumental in correlating the pathology with its clinical or physical symptoms at the beginning of the nineteenth century [9].

Due to its condition as an obligate pathogen of *Homo sapiens*, Mtb has been evolving with humanity since its origin, and when we examine the historical references it appears to have adapted to all kinds of cultural changes, which have allowed its sustainable evolution despite its incredibly mortal capacity [8]. ‘Ancient’ Mtb lineages appeared by 70,000 BCE, at a time when humans were organized into small tribes of hunter–gatherers of around 50 individuals living in Africa, with an effective population limited to a million people, who then started to expand towards the east in the process known as the second out-of-Africa [3,10,11]. This represents a very low population density [12], of around 30 people per 100 km^2^, which challenges the permanence of many infectious diseases, especially those that are human obligate parasites [10]. This low density and sustained ‘non-growing’ condition for hundreds of thousands of years, around 0.003% annually [13], was the consequence of a nomadic lifestyle where child-rearing carried a high cost, for both transportation and breeding, so it is estimated that each woman could only raise one child every 4 years [14]. Even when child mortality was high (27%), it was estimated that 46% of children died before the age of 15 [15]; these people had good quality of life, which resulted in a life expectancy of around 33 years [14,16], because once they reached the age of 15, 67% of them lived to an age of 45 or older [17].

Interestingly, modern Mtb lineages (2 to 4) appeared by 46,000 BCE, at the time of increasing expansion, progressively displacing the ancients ones (1, 5 and 6). In fact, nowadays ancient strains are detected in limited regional areas, i.e., West (lineages 5 and 6) and East Africa, the Philippines and Indian Ocean Rim (lineage 1) [1,18]. Modern lineages were less aggressive, as they were less proinflammatory, but having a greater capacity to disseminate through aerosols [19], and it is the one that finally became predominant in humans [20]. Mtb expansion was fuelled by the Neolithic revolution and the resulting explosive population growth, of around 0.1% annually, thanks to the progressive change towards sedentary life and faming-based activities, which led to higher birth rates (about one child raised every two years), but with a lower quality of life due to the impact of social inequalities [21,22], harder work duties and a less varied diet, which caused a reduction in life expectancy [16,23]. The greater dissemination capacity of modern lineages was markedly favored in the regions with a higher population growth. This can explain why the ancient ones can be.

Several models have been set up to try to improve understanding of the origin of Mtb infection in such a low-density population. One of the initial ones is based on the hypothesis that the mechanism of infection of Mtb was originally based on a late progression towards active disease (i.e., prolonged latency) of more than one generation, with the possibility that younger and more susceptible individuals became infected [24]. Adapting this criterion, Zheng et al. [25] built a model of TB transmission [26,27] using a population of 100 individuals. They concluded that to sustain TB, Mtb would have had to have a progression to disease of up to 50%, which clearly exceeds the value of 5–10% accepted nowadays [28].

Recently, we developed a susceptible–exposed–infectious–recovered (SEIR) model, which demonstrated the extraordinary impact of Mtb, causing the extinction of infected groups [19]. This could only be overcome by an unprecedented population increase attributable to annual population growth rates of 1% and 2.6% instead of the accepted 0.003% and 0.1% for the Paleolithic and Neolithic ages, respectively [13]. The study had major drawbacks, as it required a dramatic re-evaluation of the population growth parameters in prehistory.

Recent data based on the precise determination of mortality and self-cure in the natural history of TB in the prechemotherapy era have obliged us to modify this model. In this study, the authors were able to better distinguish the prognosis of smear-positive (SP) and smear-negative (SN) patients as a sign of TB severity [29]. This study shows a dramatic difference between the two forms and allows us to better explore the trade-offs that made this coevolution possible. Severe forms (SP) had an annual mortality ratio of 0.389 and self-cure ratio of 0.250, while mild forms (SN) had values of 0.025 and 0.125, respectively, reducing the mortality rate by 15-fold. Taking the TB disease spectrum into account, the impact of mild forms is paramount, as it drastically reduces mortality but maintains the possibility of disseminating the infection, albeit at a lower ratio. It is known that patients with SP have ten times higher levels of bacilli in their sputum than those with SN [30].

From this study, we hypothesized that SN patients might be the clue to reconciling the scenario of a lo- density ‘non-growing’ population with the origin and maintenance of TB. The main objective of the present article was to evaluate the SN proportion (p) needed to maintain the consensual annual population growth rates established in Paleolithic and Neolithic societies, and also to determine the impacts of ancient and modern Mtb lineages. Thus, we built a new SEIR model (TBSpectr), in which we distinguished the two clinical forms and allocated them according to a smear-negative proportion ‘p’, depending on the health status of the host, which determined the induction of one clinical form or the other.

Our work suggests that Mtb and modern humans were able to coevolve thanks to the presence of SN lesions, due to the better health status present before the Neolithic revolution. It was precisely the deteriorating health in that period that led to the increase in SP forms, even when the predominant Mtb lineages (modern ones) were less virulent. This highlights the character of TB as a poverty-related disease, and its greater impact on socially depressed sectors of the population. Additionally, the benign nature of its origin makes Mtb a highly human-adapted pathogen, which will be difficult to eradicate using the current diagnostic methods.

## 2. Results

### 2.1. Mtb Infection in the Paleolithic Was Possible Thanks to High Smear-Negative Proportion (p) Values

Adjustments of the continuous TBSpectr model (Figure 1) to the currently accepted human population growth rates in the Paleolithic (0.0003%/year) and Neolithic (0.1%/year) (Figure 2) gave us the relation between natality (λ) and the smear-negative proportion (p), which determined the allocation towards SN TB forms. We considered the demographic parameters for each period of time and Mtb lineage (Table 1). Once obtained, we established the natality value (λ) according to the annual population growth rate that was able to fit both Mtb lineages in the Paleolithic (0.032) and Neolithic (0.044) periods. This allowed us to determine the value for the smear-negative proportion (p) corresponding to each Mtb lineage for each cultural period. Thus, the *p* values were higher in the Paleolithic (0.488), corresponding to the better health status, which was then increased by the emergence of the modern lineages (0.677), which also corresponded to its lower virulence. The Neolithic period, due to the higher population growth, allowed the clinical forms to worsen dramatically according to the *p* values, corresponding to a poorer health status, which decreased in both modern and ancient lineages to 0.263 and 0.096, respectively.

We also looked at the number of children of 15 or older (Figure 2B), as the number of fertile individuals available is an interesting factor that contributes to population growth. In this case, the difference between the two periods of time was not great (2.11 vs. 2.33), indicating a higher mortality of children less than 15 years old in the Neolithic.

### 2.2. Ancient Lineages Had to Be More Virulent in the Paleolithic in Order to Sustain Mtb Infection but They Offered a Better Recovery Rate

Figure 3 shows the dynamics of the different compartments studied according to the smear-negative proportion (p). It is important to note the parabolic evolution vs. the exponential decline in SN and SP infectious cases, respectively. This indicates the need for the initial Mtb strains, belonging to ancient lineages, to have greater virulence than the modern ones; otherwise they would contribute to decrease the ‘p’ value, which would lead to a dramatic reduction in SN lesions, leading to the clearance of Mtb. Another interesting point is that the percentage of SP infectious cases relies mainly on the historical period studied, meaning that the lowering of health status in the Neolithic was responsible for the marked increase in the severity of TB. The better health status is important in the Paleolithic when looking at the percentages of recovered and infectious SN cases, as it shows a wider gap in the Paleolithic than in the Neolithic. This means that even when ancient lineages were more virulent, because they had originated in the Paleolithic, they were well tolerated. On the other hand, modern lineages increased the percentages of people exposed, regardless of the historical period studied, confirming the greater capacity for dissemination.

### 2.3. The SN Recovered Cases Were Crucial for Maintaining Mtb Infection in the Paleolithic While the Neolithic Era Was Marked by the Arrival of SP Lesions

The evolution of the people in the different compartments over a thousand years using the continuous TBSpectr model to find the stationary distribution (Figure 4) shows that the SN recovered compartment was the most important one for ensuring the persistence of Mtb infection in the Paleolithic, while also maintaining a reduced percentage of infectious people with SN lesions. The impact of SP infectious cases was minimal in this period. The emergence of modern lineages led to a slight increase in these values, making the SP infectious values even more negligible, but notably increasing the number of people exposed, which increased to reach a similar percentage to that of the group of susceptible people.

This scenario changes radically in the Neolithic period when SP infectious cases emerge dramatically, as shown in Figure 3. This is remarkable in the case of infections caused by ancient lineages. In the case of the modern ones, this emergence is influenced by the higher dissemination capacity that increases both SP and SN infectious cases together with the exposed ones, causing for the first time a clear dip in the susceptible population. Interestingly, at stationary equilibrium, both SN and SP infectious cases become similar. However, in the end the impact of Mtb infection only stops after 100 years, during the faster growth of the global population shown in the Neolithic, which can be seen by looking at the larger number of people in all compartments after the 1000-year period and explains the predominance of modern lineages (Appendix A).

### 2.4. Persistence of Mtb Infection Required Interrelations among Hunter–Gatherer Groups in the Paleolithic While the Risk of Human Extinction Was Negligible

When examining the discrete TBSpectr model to evaluate the probability of Mtb infection clearance (Figure 5), it is clear that regardless of the lineage, its survival would be impossible in reduced human groups. Data show that a minimum of a 1000-person community was necessary to maintain this during the Paleolithic, and slightly less under the infection with modern lineages. This means that the groups of 50 hunter–gatherers had to interrelate, otherwise the infection would have disappeared. This factor was less important in the Neolithic due to the higher population growth rate. On the other hand, data on the capacity of Mtb infection to cause the extinction of humankind (Figure 6) show that this was very low, at roughly around 0.2% after 500 years of coevolution in a limited group of 50 people.

### 2.5. The Number of People Infected Per Case and Fast Progression to Disease Are the Most Important Factors in Maintaining TB in Existence

A sensitivity analysis (Figure 7) using a range of values per parameter shown in Table 2 was used to analyse the influence of the different factors that define the TBSpectr model. The most important ones are precisely those that we have seen as differentiating ancient and modern Mtb lineages, i.e., the people infected per case/year (e) and rapid progression towards active TB (f), as these are the determinants for increasing the numbers in all compartments of infected people. In this regard, the immunity factor (i) seems to have the same influence, but in this case has to be read inversely, as the lower the value, the higher protection. It was of interest to confirm that the increase in the smear-negative proportion (p) increased the number of SN cases and reduced the SP ones, as expected. The reduction in infected cases by the higher mortality caused by TB (*μ*_TB,sp_) was also expected, as it reduces the chances of infection, thereby causing an increase in the susceptible compartment. The increase in natality (λ) linked to the increase in natural mortality (*μ*) was also expected, as it is precisely what happened in the Neolithic. It is interesting to note that the increase in the annual population growth rate (gr) reduced the number of exposed compartments while decreasing the SP infectious one (I_sp_). Also noteworthy is the positive correlation between the cure of SP infectious people (c_sp_) and the number of SN infectious people, showing an obvious interrelation between the two compartments. Finally, it is also interesting to observe the corroboration of how the increase in the charge value (k) raises the likelihood of SN infectious cases developing from recovered cases.

We used a heatmap to show the Partial Rank Correlation Coefficient on the TBSpectr discrete model after analyzing 1000 simulations. The influence of the evolution was shown in the relationships between the infected people per case/year (e), fast progression (*f*), immunity (i), bacillary charge (k), natural mortality (μ), reactivation factor in recovered (w), annual population growth rate (gr), cure in smear-positive (c_sp_), mortality caused by TB (μ_TBsp_) and smear-negative proportion (p) in the evolution of clearance in groups of 100 (CL100) and 1000 (CL1000) people, extinction in groups of 100 (EX100) and 1000 (EX1000) people and the equilibrium fractions of susceptible (S), exposed (E), infectious smear-negative (I_sn_), infectious smear-positive (I_sp_), recovered smear-negative (R_sn_) and recovered smear-positive (R_sp_) compartments.

## 3. Discussion

The TBSpectr model is a very significant correction of our previous model [19], which linked the origin of TB to an unprecedent population increase (×20 times in 100 years). In that case we based our assumptions on the data from a systematic review of the evolution of TB cases from the prechemotherapy era [41]. Recent data from Ragonnet et al. [29] led us to reconsider our previous work, as the authors were able to enrich that review and differentiate the evolution of SP and SN-TB separately. In our case, the model has a major flaw, as it does not consider the interrelation between the two forms of TB, but it better represents the TB spectrum to the point that it drastically changes our perception of its origin.

Following this change, our model makes an even stronger case for how coevolution between Mtb and humanity emerged around 70,000 years BCE [1]. This hypothesis is based on molecular clock calibrations that are constantly revisited [42]. Furthermore, other methodologies led to the hypothesis that its origin is linked with the control of fire, at around 300,000 to 400,000 BCE [43]. In fact, there is one report that claims the presence of *Leptomeningitis tuberculosa* in the endocranial surface of a hominid fossil with an estimated age of around 500,000 BCE in Kocabaş (Turkey) [44]. Although the interpretation of this finding has been somewhat controversial [45], it may be hypothesized that before a complete settlement as a human obligate parasite, the *Mycobacterium* species ‘attempted’ several times to become the most recent common ancestor (MRCA) of the Mtb complex that we identify nowadays. In this sense, Gutierrez et al. identified a very ancient ancestor (3,000,000 BCE) linked to the smooth tubercle bacilli (*M. canettii)* strains, which are still isolated from human TB today [46]. This position is supported by looking at the evolution of the lipid composition of the mycobacterial cell wall. In this regard, Jankute et al. have theorized the origin of Mtb as a progressive increase in the hydrophobicity of the cell wall. Thus, the origin of Mtb complex would be *M. kansasii*, with a smooth morphology—a hydrophilic, environmental mycobacterium that acts as an opportunistic pathogen of several mammals, including humans. With the progressive loss of the polar sugars together with the acquisition of apolar ones, there was an evolution towards *M. canettii* and *canetti/tuberculosis* species to finally become the Mtb complex MRCA and a human obligate pathogen [47].

Overall, this means that Mtb is the result of thousands of years of evolution of environmental mycobacteria, living in cell-free media (water and dust) or colonizing free living amoebas [48,49], and moving to colonize the ‘pulmonary amoebas’, which we can consider the alveolar macrophages, to finally becoming an obligate parasite thanks to its capacity to disseminate through aerosols [47]. However, originally this parasitization had to be sustainable in the context of a low-density human population, based in small tribal groups with a necessary interrelation between them, as demonstrated by other authors [50], to avoid clearance of the infection. Our data indicate that even by developing mild SN TB forms with low dissemination capacity, the obligate parasitization was possible. In fact, recent data on molecular epidemiology support a greater impact of subclinical TB cases than expected [51], challenging the status of the current diagnostic methodology if we are to finally eradicate this infection. In fact, looking at the countries with the lowest TB incidence rates, it appears that there is a long-lasting persistent low incidence of the infection [52], which is usually attributed to imported cases, but that we might also attribute to the original nature of Mtb infection, which tends to generate mild SN TB forms. The persistence of Mtb in our tissues would also provide some evolutive advantage to humans, for instance by increasing trained immunity and allowing people to better control acute respiratory viral infections [53]. This is a sort of mutualism, an ‘old friends’ relationship that has also been linked to several colonizing microorganisms, including environmental mycobacteria, through the balance of our inflammatory responses, which has led to the evolution of the human immune system [54,55]. In this regard, our hypothesis on the origin of the expansion of SP TB forms is supported by the reduction in standard of living, which came with the Neolithic revolution and led to increased TB severity and mortality. This took place despite the lower virulence of the newly evolved modern lineages, which had a lower inflammatory capacity but also had higher dissemination capability [19], and would not take place if the humankind would keep the healthier Paleolithic standard of living.

Our work sustains the hypothesis or idea that Mtb not only evolved to become a competent aerosol-disseminated pathogen among humans, but also gained the ability to remain sustainable among low-density populations. It was the human cultural changes in lifestyle with the Neolithic revolution that broke this balance and generated TB, the greatest killer of humankind.

Overall, our data support the concept that Mtb infection is highly adapted to persist and coevolve with humans thanks to its ability to cause long-lasting SN-TB. From a practical point of view, this means that its eradication would be very difficult because of its capacity to remain among us discretely, challenging the current diagnostic tools. It also emphasizes the poverty-related nature of the disease and the need to provide better global health status to prevent severe forms of the disease in order to be able to reduce its terrible morbidity and mortality.

The greatest strength of the model is the differentiation between SN and SP so that it is able to incorporate recent data on the determination of mortality and self-cure in the natural history of TB in the prechemotherapy era [29]. Nevertheless, the transition between SN and SP was not explored because of a lack of information about its rate, which is a limitation of the model that should be addressed in future revisions. Another limitation is that children under 15 years of age are not included in the model. Their inclusion could modify the outcome dynamics. Further work should also address the interactions between communities, which is a factor that could reveal important information concerning the maintenance of the infection.

## 4. Methodology

### 4.1. TBSpectr Model

We designed a compartmental mathematical model based on a set of differential equations to describe the dynamics of the evolution of *Mycobacterium tuberculosis* complex (MtbC) infection in the population, based on our previous study in several models [19].

The population is divided into several compartments according to their status with regards to the infection cycle: S, susceptible; E, non-infectious exposed; I_SP_, smear-positive infectious; I_SN_, smear-negative infectious; R_SP_, recovered from a smear-positive TB course; R_SN_, recovered from a smear-negative TB course. Newborn individuals appear at a rate π in the susceptible compartment. Mortality is given by μ when it is not caused by TB, $\mu_{TB,SP}$ when it is caused by SP tuberculosis and $\mu_{TB,SN}$ when it is caused by SN tuberculosis. Flows between compartments are shown in Figure 1 and given by this set of equations:(1)$$\frac{dS}{dt}=\pi +\delta E-\mu S-\beta IS$$
(2)$$\frac{dE}{dt}=\left(1-f\right)\;\beta SI-\left(\mu +\delta +i\left(a+rI\right)\right)E$$
(3)$$\frac{dI_{SP}}{dt}=f\left(1-p\right)\beta SI+i\left(1-p\right)\left(a+rI\right)E+\left(1-p\right)wi\left(a+rI\right)R-\left(\mu +\mu_{TB,SP}+c_{SP}\right)I_{SP}$$
(4)$$\frac{dI_{SN}}{dt}=fp\beta SI+ip\left(a+rI\right)E+pwi\left(a+rI\right)R-\left(\mu +\mu_{TB,SN}+c_{SN}\right)I_{SN}$$
(5)$$\frac{dR_{SP}}{dt}=c_{SP}I_{SP}-\left(\mu +wi\left(a+rI\right)\right)R_{SP}$$
(6)$$\frac{dR_{SN}}{dt}=c_{SN}I_{SN}-\left(\mu +wi\left(a+rI\right)\right)R_{SN}$$
(7)$$\frac{dN}{dt}=\frac{dS}{dt}+\frac{dE}{dt}+\frac{dI}{dt}+\frac{dR}{dt}$$
where $I=I_{SP}+I_{SN}$ and $R=R_{SP}+R_{SN}$. The different parameters that appear in the equations are the following: birth rate (π); percentage of bacillary drainage of the infection on exposed individuals (δ); non-TB-caused mortality rate per year (μ); TB transmission rate (β); percentage of fast progression of the active disease (f); reduced progression due to immunity (i); reactivation of an infection (a); risk of disease caused by a reinfection (r); TB-caused mortality rate ($\mu_{TB,SP}$); increased progression in recovered in comparison to naïve (w); smear-negative proportion (p); natural cure rate of smear-negative ($c_{SN}$); and natural cure rate of smear-positive ($c_{SP}$). Table 1 presents a summary of the parameters, the ranges of values explored and the sources, when available. The following paragraphs describe the relationship among some of them.

The transmission rate β depends on the number of new infections caused by a particular SP or SN case ($e_{SP}$ and $e_{SN}$), as well as on the proportion of each infectious compartment: $\beta =\frac{1}{N}\frac{e_{SP}I_{SP}+e_{SN}I_{SN}}{I_{SP}+I_{SN}}$ [27]. The probability of showing rapid progression of the disease during the first year post-infection or reinfection is given by *f* [36,56], while the progression to active TB once TB infection remains latent (E) or active TB naturally recovers (R) is included with the parameter a, which represents 30% of the fast progression according to the data from Sloot et al. [36]. The parameter *r* provides the risk of disease caused by reinfection as r=f β+a(1−f)β. This relationship considers that given a reinfection, its progression to active TB can be due both to fast progression (*f*) and reactivation (*a*) of the new infection [19]. We consider a protective ratio i of 0.1 (i.e., those infected people that do not develop the disease have at least 90% protection against the onset of disease) [39]. The drainage of the infection on exposed individuals is given by δ [37,38], which is assumed to be reduced by the possibility of endogenous or exogenous reactivation of the infection, defined as a and r, respectively. Thus, bacillary drainage is defined as: δ = 0.1 − *a* − *r*. Uys et al. [40] determined that recovered subjects have a seven-fold higher chance of developing disease, which is given by *w* in our model. The birth rate (π) and natality rate (λ) determine the newborn individuals π=λ(S+E+I+R). The natality parameter is adjusted using all other input parameters to fit the annual population growth rate (gr). The number of births per fertile woman and year are computed as $2\cdot \mu^{-1}\cdot \lambda$. We also included the *k* factor (0.1) to take into account the fact that SN-TB patients have ten times lower levels of bacilli in their sputum than SP patients, and that this has an impact on reducing the reactivation cases in recovered SN-TB [30].

The assumptions behind each of the equations and values have been discussed at length previously [19]. The main novelty of this update of the model is the distinction between SP and SN-TB. In this regard, we introduced the smear-negative proportion *p*, which determines the percentage of active cases that show a smear-negative course. The annual spontaneous cure ($c_{SP}$ and $c_{SN}$) and dying (μ_TB,SP_ and μ_TB,SN_) rates depend on the SP or SN nature of the disease and are given by Ragonnet et al. [29]. We did not include a transition between SN and SP, as we considered only the final clinical status and could not establish the rate of transition between them.

### 4.2. Assessment of Uncertainty and Sensitivity in the System

The uncertainty and sensitivity analysis of the TBSpectr model was performed as described in [57]. We used a Latin hypercube sampling (LHS) technique to generate 1000 different parameter sets representative of the parameter space (sampling-based method). Parameters between the values shown in Table 2 were explored. The partial rank correlation coefficient (PRCC) was computed at each time step for each of the parameters and susceptible, exposed, infected, recovered and total populations, as were the annual incidence and death rates. We also computed the final PRCC rates between input parameters and TB clearance and community extinction, using discrete resolution (see below). This analysis allowed for the assessment of the individual effect of each parameter on each outcome, with a linear ruling out of the effects of the uncertainty on the rest of the parameters.

### 4.3. Continuous and Discrete Resolution of the Models

The continuous TBSpectr model was numerically integrated with MATLAB using the Euler method, with an integration step of 1/10 years. As a result of the integration, we obtained the dynamics of each of the model’s variables.

As we discussed in our previous paper [23], the limited size of some of the communities studied suggests the suitability of exploring a discrete resolution of the model, using natural numbers to describe the variable dynamics. With such discrete resolution, which was also based on the Euler integration method, we converted each of the flows at each integration step into a natural number using Poisson random distribution. This use of randomness on the flow rounding makes it possible that two communities with the same initial conditions and model parameters diverge in their evolution. In particular, there is a chance that a certain community achieves clearance of the infection, while other communities remain affected by TB or can even become extinct because of the pathogen.

## Figures and Tables

**Figure 1 pathogens-11-00366-f001:**
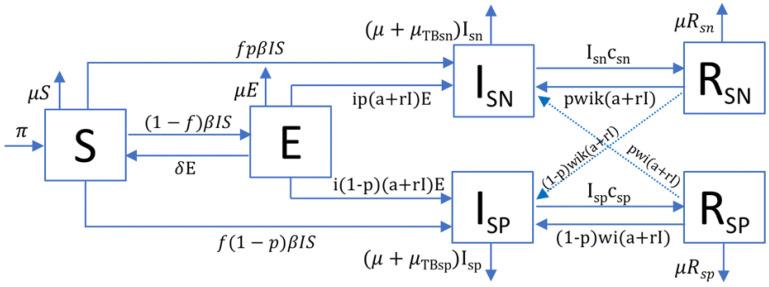
TBSpectr model. Each compartment refers to the set of individuals by disease status: susceptible, exposed, infected, recovered. New-born individuals are assumed to be susceptible. A TB infection can remain latent (**E**) or can directly develop into infectious active TB (**I**). The latent TB infection can become active through endogenous reactivation or exogenous reinfection. Patients with active TB can naturally recover (**R**), becoming non-infectious. We identified two categories in I and R, according to the spectrum of the disease, and we distinguished smear-negative (**SN**) and smear-positive (**SP**) forms of active TB. The driver for the evolution towards both forms of the disease depends on the value of a smear-negative proportion (p), which in turn depends on the health status of the host. By the same token, we have included a factor depending on the bacillary load (k) linked to smear-negative cases. Latent infected persons (**I**) can drain bacilli, lose immunity and become susceptible (**S**). Recovered persons can relapse to active TB through endogenous reactivation or exogenous reinfection. Factors included are: birth rate (π), transmission rate (β), drainage of infection (δ), fast progression (*f*), immunity (i), endogenous reactivation (a), exogenous reinfection (r), natural mortality (μ), smear-negative proportion (p), bacillary charge (k), reactivation factor in recovered (w), cure in smear-positive (c_sp_) and smear-negative (c_sn_), mortality caused by TB in smear-positive (μ_TBsp_) and smear-negative (μ_TBsn_), infectious smear-negative (I_sn_), infectious smear-positive (I_sp_), recovered smear-negative (R_sn_) and recovered smear-positive (R_sp_) compartments.

**Figure 2 pathogens-11-00366-f002:**
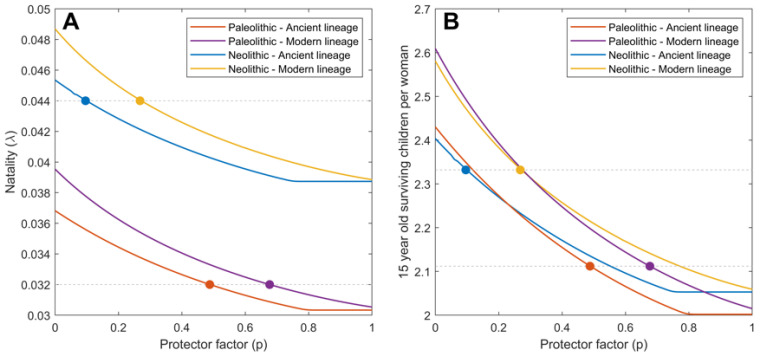
Relation between the smear-negative proportion (p) and natality. Adjustment of natality (**A**) and effective natality is understood as the number of surviving 15-year-old children per woman (**B**) using the parameters of the TBSpectr model and the accepted growth population rates for the Paleolithic (0.0003%/year) and Neolithic (0.1%/year). Colored lines represent the values for the Paleolithic period and the infection with ancient (orange) and modern (violet) variants of Mtb, as well as the Neolithic period and the infection with ancient (blue) and Modern (yellow) variants of Mtb. Dots of corresponding colors show the values for the smear-negative proportion (p) chosen in each case.

**Figure 3 pathogens-11-00366-f003:**
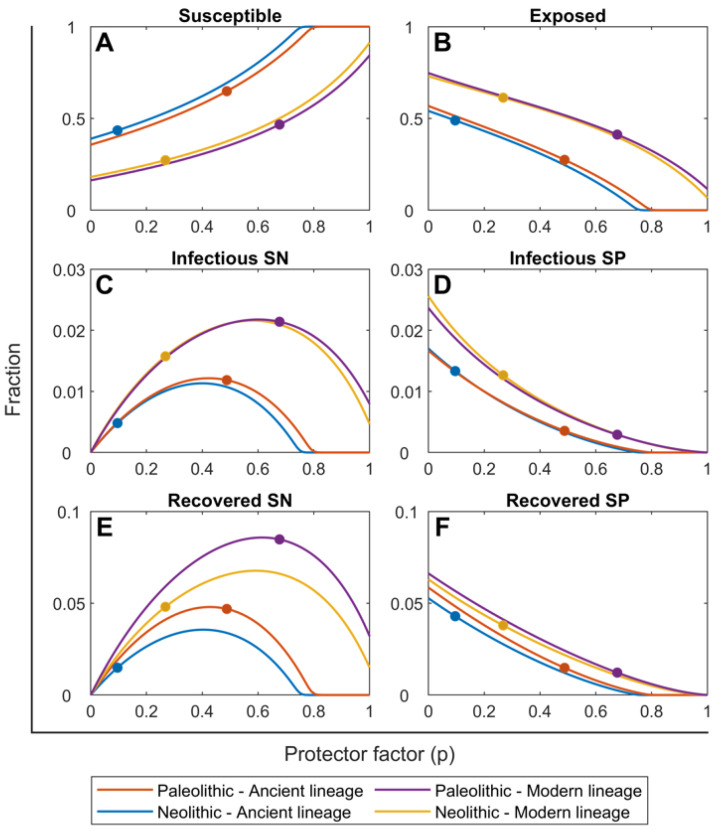
Evolution of the percentages of individuals in each SEIR compartment in relation to the smear-negative proportion (p) according to the continuous TBSPectr model. Pictures show the evolution rates obtained for susceptible (**A**), exposed (**B**), infected SN (**C**), infected SP (**D**), recovered SN (**E**) and recovered SP (**F**) cases. Fractions are independent of initial conditions. Colored lines represent the values for the Paleolithic period and the infection with a ancient (orange) and modern (violet) variants of Mtb, as well as the Neolithic period and the infection with ancient (blue) and modern (yellow) variants of Mtb. Dots of each corresponding color show the value of the smear-negative proportion (p) chosen in each case.

**Figure 4 pathogens-11-00366-f004:**
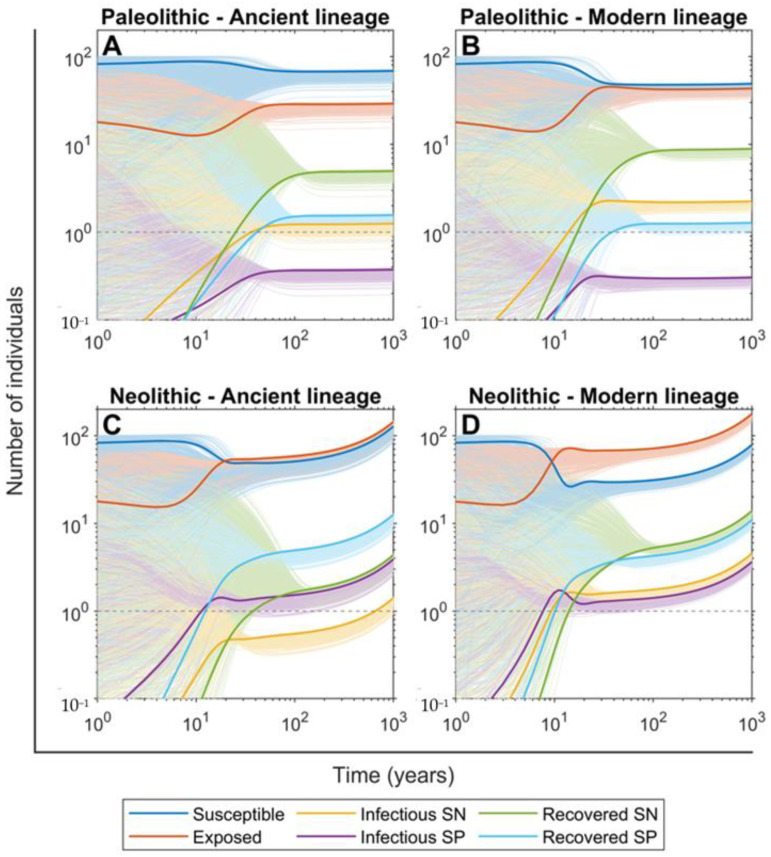
Evolution of the population in the continuous TBSpectr model towards the stationary state. Pictures show the projections of 1000 simulations in a group of 100 people with a random initial distribution among compartments (S_0_
∈ [0, 100], E_0_
∈ [0, 40], I_c0_, I_n0_, R_c0_, R_n0_
∈ [0, 100−S_0_−E_0_], the sum of all compartments is equal to 100) during 1000 years of evolution. The thick lines correspond to the central scenario with S_0_ = 80, E_0_ = 20 and other compartments starting at zero. Evolution rates are drawn for the Paleolithic period and the infection with ancient (**A**) and modern (**B**) variants of Mtb, as well as the Neolithic period and the infection with ancient (**C**) and modern (**D**) variants of Mtb. Colour lines represent the different compartments of susceptible (blue), exposed (orange), infected SN (yellow), infected SP (violet), recovered SN (green) and recovered SP (cerulean blue). For reference, there is a grey dotted horizontal line marking the presence of 1 person (10^0^log_10_).

**Figure 5 pathogens-11-00366-f005:**
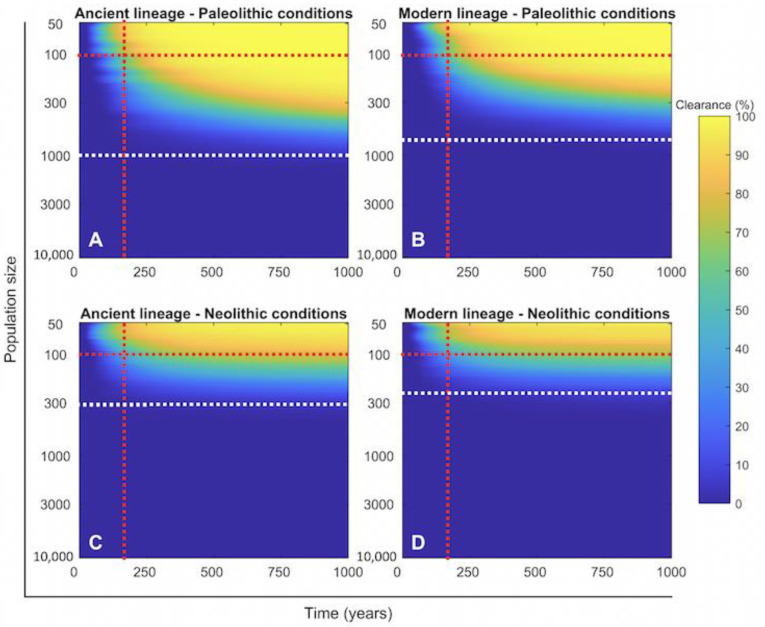
Relation between the size of the human group and the clearance of the Mtb infection. Heatmap of the end values for 1000 year’s evolution on the TBSpectr discrete model using the initial conditions in each compartment found in the equilibrium phase shown previously (Figure 4). For reference, we have included a vertical dotted red line at the time where there is a 100% clearance for the smallest population, a horizontal dotted red line to reference the evolution in a population size of 100 people and a white line at the population size where there is 0% clearance in the Paleolithic period under the infection of Mtb ancient lineages. Clearance means the lack of population in the Exposed, Infectious, and Recovered compartments. We used 1000 simulations with random initial conditions, using parameters from Table 1.

**Figure 6 pathogens-11-00366-f006:**
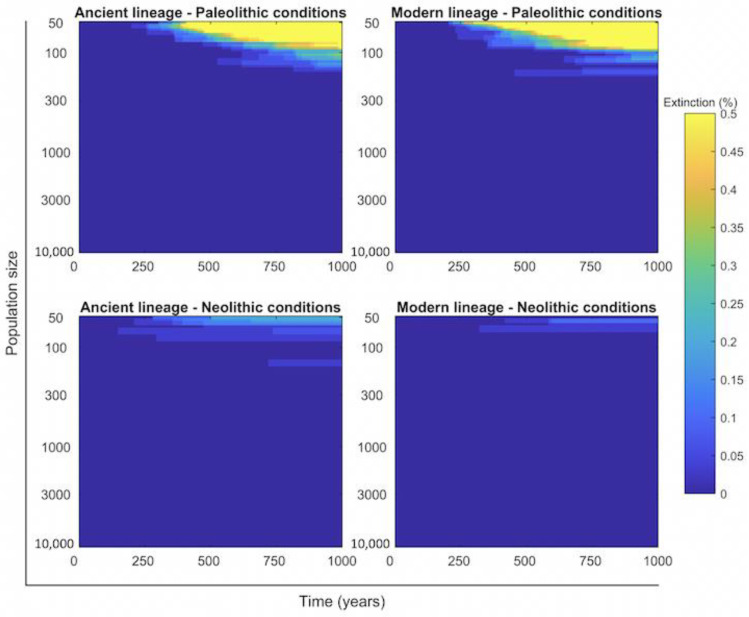
Relation of the size of the human group to the extinction of humankind. Heatmap of the end values for 1000 year’s evolution on the TBSpectr discrete model using the initial conditions in each compartment found in the equilibrium phase shown previously (Figure 4). Extinction means the disappearance of humankind in the group explored.

**Figure 7 pathogens-11-00366-f007:**
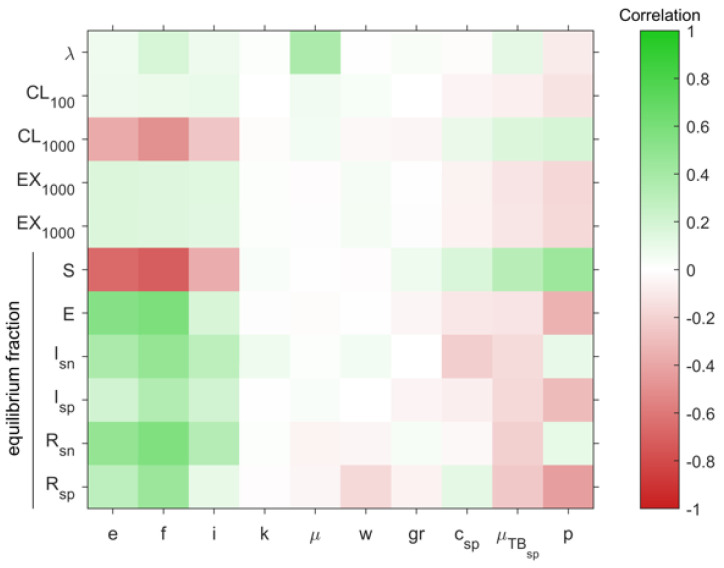
Sensitivity analysis.

**Table 1 pathogens-11-00366-t001:** Parameters and references.

Parameter	Values		Sources
	Paleolithic	Neolithic	
Annual population growth rate (gr)	0.003%	0.1%	[13,31,32]
Natality (λ)	0.032	0.044	
15 years old surviving children/women	2.11	2.33	
Natural mortality/year (μ)	1/33	1/26.5	[33,34]
Smear-negative proportion (p)	0.488 (A) 0.677 (M)	0.096 (A) 0.268 (M)	explored
Mortality/year caused by TB (μ_TB_)	0.389 (SP)/0.025 (SN)		[29]
Infected people per case/year (e)	A = 10 (SP)/1 (SN); M = 20 (SP)/2 (SN)		[30,35]
Bacillary charge (k)	0.1		[30]
Fast progression (*f*)	0.099 (A)/0.0825 (M)	0.1238 (A)/0.1031 (M)	[36]
Reactivation from infection (a)	f 0.3		
Bacillary drainage and immunity reduction (δ)	0.1-a-r		[37,38]
Reduced progression due to immunity (i)	0.1		[39]
TB natural cure (c)	0.231 (SP)/0.130 (SN)		[29]
Increased progression in recovered (w)	7		[40]

SP: Smear-positive; SN: Smear-negative; A: Ancient strain; M: Modern strain.

**Table 2 pathogens-11-00366-t002:** Sensitivity analysis.

Parameter	Paleolithic Value	Neolithic Value	Sensitivity Analysis Range %
gr	0.003%	0.1%	(0.003, 0.1)
μ	0.03030	0.03846	(0.0286, 0.04)
p	0.488 (A) 0.677 (M)	0.096 (A) 0.268 (M)	(0, 1)
$\mu_{\mathrm{TB}}$	0.389 (SP) 0.025 (SN)		(0.02, 0.4)
e	A = 10 (SP) 1 (SN); M = 20 (SP) 2 (SN)		(1, 20)
f	0.099 (A) 0.0825 (M)	0.1238 (A) 0.1031 (M)	(0, 0.13)
i	0.1		(0.05, 0.5)
c	0.231 (SP) 0.130 (SN)		(0.1, 0.4)
w	7		(1, 7)
k	0.1		(0.05, 0.2)

SP: Smear-positive; SN: Smear-negative; A: Ancient strain; M: Modern strain.

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
