# Peer review of "The Origin and Maintenance of Tuberculosis Is Explained by the Induction of Smear-Negative Disease in the Paleolithic"

_pathogens, 2022, doi:10.3390/pathogens11030366_

Round 1

Reviewer 1 Report

The model presented by Cardona et al. provides an interesting extension to that presented by Ragonnet et al., highlighting the differential contributions of SN and SP states to this pathogen's maintenance and the importance of socioeconomic conditions to the progression of the infection not only in modern humanity but also in prehistory. Among the limitations to the model that authors highlight (not accounting for transitions between SN and SP states, etc.), including considerations for pediatric infections would be of most interest, considering the higher pathogenicity of this pathogen in children and differences in infection course and outcomes. 

The study, while interesting, is somewhat difficult to read due to issues with English language and grammar throughout - editing for this is highly recommended before publication. 

Author Response

An english mother tongue writer has reviewed again the paper to properly edit english language and grammar. We hope this will be more readable now.

Reviewer 2 Report

The authors have provided a mathematical model to hypothesize how Mycobacterium tuberculosis (Mtb), a deadly pathogen that kills about 2 million people worldwide annually has its origin in Middle Paleolithic. Their model is based on molecular clock calibrations. The authors had previous published a model to trace the origin of the pathogen termed as Susceptible-Exposed-Infectious-Recovered (SEIR) that linked the origin of the tuberculosis to an unprecedented increase in the population growth rate. However, the model had flaws because it did not consider the difference between virulent Smear Positive (SP) form of tuberculosis that originated in the Neolithic Age to that of comparative less virulent variant Smear Negative forms of TB that played a key role in persistence within the human populations during the Paleolithic Age as the Smear Negative form have much less bacterial load on the infected individual. The authors suggest that Mtb infection was possible in the Paleolithic era mainly due to the Smear negative form. The recovered cases from Smear Negative form of TB acted as a reservoir for maintenance of the pathogen during the Paleolithic era. However, the Neolithic era is marked by the advent of the Smear Positive variant of the pathogen. The model suggests that for Mtb to persist in the human population, it must be present in at least “1000” individuals leading to the clue that the hunter gatherers group that comprises typically of 50 individuals must be able come in close contact with each other for viable transmission of the disease. The authors have conducted sensitivity analysis to show that number of people infected per case and the fast progression of the disease are the two key attributes that kept TB persistent in the human population. The author’s work is also supported by previously published work that focused on evolution of lipid composition of mycobacterial cell wall. These works elucidate that there was marked reduction of the polar sugars in the cell wall composition of Mycobacterium tuberculosis and there was a greater representation of the non-polar components that resemble the cell wall characteristics of the present day Mtb Complex. Overall, the paper has presented the results clearly and should be of interest to people studying bacterial pathogenesis.

However, I have noticed couple of typos in the manuscript. The authors should recheck the manuscript.

Minor comments:

It was not clear from the paper how the modern and the ancient lineages of Mtb played differential role among populations that spread to different regions when they spread out of Africa. Is there any paper/study that showed that one form of Mtb was more prevalent among a certain geographic location compared to other and how did migration affect the two forms of Mtb infection?

In Figure 4

    1. Why is there a dip in the susceptible category in the Neolithic-Modern lineage? The author should comment on that.
    2. Are these graphs normalized against the human growth rate as the human growth rate should tremendous increase with time?

As the human populations grew, so did the growth rate. How did the human growth rate affect the dynamics between the two forms of Mtb infection? The authors should comment on that.

Author Response

1.-We have reviewed the paper again in order to improve english language and grammar.

2.-We have included in the introduction the geographic areas where ancient lineages are more prevalent: "Interestingly, modern Mtb lineages (2 to 4) appeared by 46,000 BCE, at the time of increasing expansion, progressively displacing the ancients ones (1, 5 and 6). In fact, nowadays ancient strains are detected in limited regional areas, i.e. West (lineages 5 and 6) and East Africa, the Philippines and Indian Ocean Rim (lineage 1)(1,18)". We have cited the papers of Comas et al (1) and Gagneux and Coscolla (18).

3.-Regarding the dip in the susceptible compartment, we have included the following sentence: "In the case of the modern ones this emergence is influenced by it’s higher dissemination capacity that increases both SP and SN infectious cases together with the exposed ones, causing for the first time a clear dip in the susceptible population. Interestingly, at the stationary equilibrium, both SN and SP infectious cases become similar".

4.-In the figure 4 we have not normalized the data. We have used the specific growing rates for the Paleolithic and the Neolithic displayed in Table 1.

5.-We have included a comment on that in the 2.5. chapter: "What is interesting to note is that the increase in the annual population growth rate (gr) reduced the number of exposed compartment while decreasing the SP infectious one (ISP)".